# TMT-Based Comparative Proteomic Analysis of the Spermatozoa of Buck (Capra hircus) and Ram (Ovis aries)

**DOI:** 10.3390/genes14050973

**Published:** 2023-04-25

**Authors:** Chunhuan Ren, Yale Chen, Jun Tang, Penghui Wang, Yan Zhang, Chunyan Li, Zijun Zhang, Xiao Cheng

**Affiliations:** 1College of Animal Science and Technology, Anhui Agricultural University, Hefei 230036, China; 2Yunnan Academy of Animal Husbandry Veterinary Sciences, Kunming 650224, China; 3Modern Agricultural Technology Cooperation and Popularization Center of Dingyuan County, Chuzhou 233200, China

**Keywords:** bucks and rams, spermatozoa, proteomics, tandem mass tag

## Abstract

Spermatozoa are unique cells that carry a library of proteins that regulate the functions of molecules to achieve functional capabilities. Currently, large amounts of protein have been identified in spermatozoa from different species using proteomic approaches. However, the proteome characteristics and regulatory mechanisms of spermatozoa in bucks versus rams have not been fully unraveled. In this study, we performed a tandem mass tag (TMT)-labeled quantitative proteomic analysis to investigate the protein profiles in the spermatozoa of buck (Capra hircus) and ram (Ovis aries), two important economic livestock species with different fertility potentials. Overall, 2644 proteins were identified and quantified via this approach. Thus, 279 differentially abundant proteins (DAPs) were filtered with a *p*-value < 0.05, and a quantitative ratio of >2.0 or <0.5 (fold change, FC) in bucks versus rams, wherein 153 were upregulated and 126 were downregulated. Bioinformatics analysis revealed that these DAPs were mainly localized in the mitochondria, extracellular and in the nucleus, and were involved in sperm motility, membrane components, oxidoreductase activity, endopeptidase complex and proteasome-mediated ubiquitin-dependent protein catabolism. Specifically, partial DAPs, such as heat shock protein 90 α family class a member 1 (HSP90AA1), adenosine triphosphate citrate lyase (ACLY), proteasome 26S subunit and non-ATPase 4 (PSMD4), act as “cross-talk” nodes in protein–protein networks as key intermediates or enzymes, which are mainly involved in responses to stimuli, catalytic activity and molecular function regulator pathways that are strictly related to spermatozoa function. The results of our study offer valuable insights into the molecular mechanisms of ram spermatozoa function, and also promote an efficient spermatozoa utilization link to fertility or specific biotechnologies for bucks and rams.

## 1. Introduction

Spermatozoa are specific cells with inactive transcription that need to undergo a series of physiological and biochemical changes in the epididymis to mature, and to subsequently capacitate in the female genital tract with final fertilization potential. Male fertility is of great significance in the animal natural mating and breeding industry, and in assisted breeding. Semen freezing and artificial insemination (AI) are available procedures to rapidly disseminate elite genetic material over a large geographic area in several domestic animals [1]. It should be noted that the ultrastructure, biochemistry and function of spermatozoa suffer damage during these processes, reducing the viability [2,3]. There are differences in sperm morphology and concentration among species [4,5], which indicate various abilities to undergo cryopreservation, capacitation and fertility [6]. Therefore, a more in-depth study of sperm biology is urgently needed to further improve the production efficiency of superior species through AI.

Proteomic strategies are widely used to solve biological problems, and are a common and effective method for protein characterization [7]. Tandem mass tags (TMT) technology is an in vitro peptide labeling technique developed by Thermo Scientific, Inc., for the identification and quantification of proteins in different samples. The technique can use up to 11 stable isotope tags to compare the relative amounts of proteins in up to 10 different samples simultaneously, and can be used to label almost any peptide or protein sample [8,9]. Recently, this technique has been widely used to identify a cascade of sperm-specific proteins involved in cell-cell adhesion, motility performance and plasma membrane changes in humans [10,11] and mice [12]. We have studied the high- and low-motility spermatozoa of rams, detected a total of 150 differential proteins using a TMT-based method, and obtained phosphatidylethanolamine binding protein 4 as a potential marker for sperm motility [13]. Moreover, by comparing proteomic differences between fresh and cryopreserved ram sperm before and after energization, Peris-Frau et al. (2019) found that cryopreservation accelerated the apoptotic stress response and redox processes in sperm, and that sperm energization increased biological processes related to signaling, metabolism, motility and reproduction. Furthermore, based on the TMT technique, researchers have acquired 26 critical proteins in buck spermatozoa that are mainly involved in apoptosis regulation and oxidative phosphorylation during liquid storage [14,15]. All of these findings emphasize the importance of the sperm proteome for exploring functional proteins involved in cellular regulation mechanisms, which serve effective sperm fertility.

Bucks and rams are two important species of domestic artiodactyls that can provide meat, milk, wool or skin for our society. With the rapid development of modern intensified agriculture, the commercial production of highly productive bucks and rams derived from artificial selection and breeding is increasing to meet the market demand for such animal products. Their sperm will vary according to specific characteristics such as size, shape, concentration and microenvironmental composition. However, due to a specific phospholipase contained in the seminal plasma of bucks, this influences the viability of sperm by interacting with the milk or egg yolk [16]. Ram sperm nuclei have greater width than those in bucks, while being clearly less elliptical and elongated with greater regularity [17]. These characteristics affect the fertility differences in spermatozoa between the two species. In light of this, we characterized and compared the spermatozoa proteins of bucks and rams using the TMT-based approach, to better investigate the underlying molecular mechanism of their sperm characteristics and functional roles; this research provides a reference for assisted breeding of the two species.

## 2. Materials and Methods

### 2.1. Experimental Design and Workflow

This study focused on the differences in sperm proteins between bucks and rams, using a TMT proteomic quantification technology. The experimental design and workflow are shown in Figure 1. Protein identification was performed using protein extraction followed by TMT labeling and mass spectrometry, and the results were subsequently analyzed bioinformatically.

### 2.2. Animals and Sample Collection

All experimental procedures were approved by the Institutional Animal Care and Use Committee (IACUC), Anhui Agricultural University, Hefei, China (approval number AHAUB2022009). Adult Anhui white bucks (n = 9) and Hu rams (n = 9) at 2 years of age were selected. All of the animals were free to eat forage, access water, and exercise sufficiently in two farms of the Anxin Animal Husbandry Development Co., Ltd. (Bozhou, China). Each animal was fed 2 kg of fresh grass, 0.8 kg of hay and 0.5 kg of mixed concentrate (half maize and half soya meal) per day during the period of semen collection. In December, fresh ejaculate (2 per ram) was collected from 18 rams through a pseudovagina after 4 days of abstinence induced by an estrous female. The ejaculate samples were immediately assessed for motility using a computer-assisted sperm analysis system (CASAS, Hamilton Thorne, Beverly, MA, USA); spermatozoa with a concentration over 3 × 10^9^/mL and the motility greater than 75% were accepted [15]. Then, semen samples per species were divided into 3 groups randomly (n = 3) (we defined bucks as group A, rams as group B).

Two group samples (n = 3 × 2) were centrifuged at 3000× *g* for 15 min at 4 °C, and the supernatants were discarded. The sperm pellets were washed and separately three times in 1× phosphate-buffered solution (1 × PBS; Gibco, Thermo Scientific, Wilmington, DE, USA) by centrifugation (3000× *g*, 5 min, 4 °C). Then, the sperm pellets were kept in liquid nitrogen until the next step [13].

### 2.3. Protein Extraction and Digestion

Each 200–250 μL sperm pellet was resuspended in lysis buffer (8 M urea [Sigma, Darmstadt, Germany], 2 mM EDTA [Sigma], 10 mM dithiothreitol [DTT, Sigma] and 1% protease inhibitor cocktail [Calbiochem, Darmstadt, Germany]) and incubated for 2 min, then sonicated three times on ice using a high-intensity ultrasonic processor (HX-1000, Shanghai, China). The remaining debris was removed after centrifugation (20,000× *g*, 4 °C, 10 min). Finally, using 15% trichloroacetic acid (Thermo Fisher Scientific, Waltham, MA, USA) for 2 h at −20 °C, the proteins were allowed to form a precipitate. The supernatant was discarded after centrifugation at 4 °C for 10 min, and the remaining precipitate was washed with cold acetone three times. The protein concentration was determined with a 2-D Quant kit (GE Healthcare, Little Chalfont, UK), according to the manufacturer’s instructions (Appendix A). The 5–8 μL extracted protein was separated via SDS-PAGE, and the protein profile from each sample presented on the gel (Appendix A) [11].

During digestion, 100 μg of protein from each sample was reduced with 100 mM DTT at 56 °C for 30 min, followed by the addition of 20 mM iodoacetamide (IAA, Sigma, Darmstadt, Germany) at room temperature (RT) while in the dark with alkylating solution for 45 min. Then, the protein sample was diluted by adding 100 mM triethylammonium bicarbonate (TEAB, Sigma, Darmstadt, Germany) to a urea concentration below 2 M. Finally, each protein sample was digested with trypsin at a mass ratio of 1:50 (trypsin:protein) for the first digestion overnight for 16–18 h, and at 1:100 (trypsin:protein) for the second 4 h digestion.

### 2.4. TMT Labeling

After trypsin digestion, the peptides were desalted with a Strata X C18 SPE column (Phenomenex) and vacuum dried. Subsequently, the peptides were reconstituted in 0.5 M TEAB and processed according to the manufacturer’s protocol for the 6-plex TMT kit (ThermoFisher Scientific, Waltham, MA, USA). Briefly, one unit of TMT reagent (defined as the amount of reagent required to label 100 μg of proteins) was equilibrated at RT; 100 μg of each sample was resuspended in 24 μL of anhydrous acetonitrile (ACN), and TMT reagent was added to the peptides dissolved in 0.5 M TEAB. After 2 h at RT, 8 μL of 5% hydroxylamine (*w*:*v*) was added and incubated for 15 min. The samples were then combined, desalted and dried via vacuum centrifugation [11,18].

### 2.5. High-Performance Liquid Chromatography (HPLC) Fractionation of Peptides

The dried peptides were then fractionated into fractions using a high-pH reverse-phase HPLC system fitted with an Agilent 300Extend C18 column (5 μm particles, 4.6 mm ID, 250 mm length). In brief, the peptides were first separated with a gradient of 2% to 60% acetonitrile in 10 mM ammonium bicarbonate (pH 10) over 80 min into 80 fractions. Then, the peptides were combined into 18 fractions and dried with vacuum centrifugation.

### 2.6. Liquid Chromatography Coupled with Tandem mass Spectrometry (LC–MS/MS) Analysis

The peptides were dissolved in 0.1% solvent A (formic acid, FA), directly loaded onto a reversed-phase pre-column (Thermo Scientific, Acclaim PepMap 100, USA). Peptide separation was performed using a reversed-phase analytical column (Thermo Scientific, Acclaim PepMap RSLC). The gradient was increased with solvent B (0.1% FA, 90% ACN) from 8% to 26% in 22 min, from 26% to 40% in 12 min, and increased to 80% in 3 min to remain at 80% for the last 3 min. This was carried out at a constant flow rate of 400 nL/min using an EASY-nLC 1000 UPLC system. The peptides were then analyzed on a Q Exactive^TM^ plus hybrid quadrupole-orbitrap mass spectrometer (ThermoFisher Scientific, USA).

The peptides were analyzed via the Q ExactiveTM plus (Thermo Scientific) with a positive ion model and data-dependent acquisition. The resolution of the MS scan was 70,000, and the ion fragment was 17,500. Based on the MS scan, the top 20 precursor ions were selected to fragment with 30 s of dynamic exclusion. The electrospray voltage applied was 2.0 kV. The MS/MS spectra were generated using automatic gain control (AGC) to prevent overfilling of the Orbitrap and accumulation of 5E4 ions. For the MS scans, the *m*/*z* scan range was set from 350 to 1800. The first fixed mass was set at 100 *m*/*z*.

### 2.7. Identification and Quantification of Proteins

The raw MS/MS data were processed using MaxQuant with the integrated Andromeda search engine (v. 1.5.2.8). The data were compared against a combined database of Capra hircus, Ovis aries and Bos taurus. The detailed parameter settings were as follows: cleavage enzyme = trypsin, maximum number of missing cut points = 2, precursor ion mass error = 10 ppm, fragment ion mass error = 0.02 Da, fixed modifications = carbamidomethyl (C), TMT-6 plex (N-term), TMT-6 plex (K), variable modifications = oxidation (M), TMT-6 plex (Y), minimum peptide length = 7, false discovery rate (FDR) for protein, peptide and modification site ≤ 1%. For the quantification method, TMT-6-plex was selected. All of the other parameters in MaxQuant were set to default values. The relative protein abundance ratios between the two groups were calculated from TMT reagent reporter ion intensities from higher energy collision dissociation spectra, according to the Libra algorithm of the trans-proteomic pipeline. Proteins in each sample were normalized against the pooled sample in the corresponding experiment for statistical comparison. The means and standard deviations were calculated for each protein across groups [11]. Enrichment of differentially abundant proteins (DAPs) was tested against all identified proteins using a two-tailed Fisher’s exact test. Standard FDR control methods were used to correct for multiple hypothesis testing. A corrected *p*-value < 0.05 was considered significant.

### 2.8. Western Blotting Verification

Western blotting was performed to verify the proteomic data. The samples were transferred onto PVDF membranes (ipvh00010) after SDS-PAGE (10%) separation with the Mini trans-blot system (Bio-Rad), requiring 60 ug per sample. After the transfer, the membranes were blocked in T-TBS (contained 5% (*w*/*v*) skim milk powder and 0.1% (*v*/*v*) Tween 20) for 1 h at RT, and then incubated overnight at 4 °C with each of the following primary antibodies: rabbit anti-PSMD3 (ab140440, 1:500, UK), rabbit anti-DNAJB13 (ab185301, 1:500), rabbit anti-ACLY (ab40793, 1:1000), rabbit anti-DLST (ab187692, 1:1000) and mouse anti-HSP90AA1 (ab79849, 1:1000). After incubation, the membranes were washed in T-TBS four times for 5 min each, and then incubated with goat-anti-rabbit IgG (H + L) secondary antibody (Thermo Pierce, No.31460, 1:5000, Waltham, MA, USA) or goat-anti-mouse IgG (H + L) secondary antibody (Thermo Pierce, No.31431, 1:5000) for 1 h at RT. After incubation, the PVDF membranes were shaken at low speed for 10 min on a shaker and washed repeatedly 3–5 times. To detect the specific signal by chemiluminescence (ECL, Thermo Pierce), the luminescent solution (liquid A, liquid B) was configured 1:1 (pay attention to avoid light), and the appropriate amount of luminescent solution was pipetted to cover the PVDF membrane, exposed on the ECL luminometer, and the images were acquired; the blotted bands were identified using Image Pro plus 6.0 software [18].

### 2.9. Bioinformatics Analysis

In this study, bioinformatics analyses were mainly performed, such as protein subcellular localization (PSL), hierarchical clustering, gene ontology (GO) annotation, KEGG (Kyoto Encyclopedia of Genes and Genomes) annotation and interaction of DAPs analyses. PSL was performed via wolfpsort (http://wolfpsort.seq.cbrc.jp/, accessed on 6 January 2020) to determine the subcellular location where the protein or expression product should be located. Hierarchical clustering was performed based on the “gplots” R-package (v.3.6.0). To explore the functional significance of DAPs, GO and pathway enrichment analysis was performed. The proteins were classified into three categories after annotation by GO: biological processes, cellular components and molecular functions. The structural domains of the identified proteins were annotated using the tongInterPro database [19]. The KEGG database was used for KEGG annotation. The enrichment analysis was performed using a two-tailed Fisher’s exact test, and corrected *p* values < 0.05 were considered significant [20]. The webtool STRING 11.0 (http://string-db.org, accessed on 10 January 2020)) was used to analyze protein–protein interactions (PBIs), selecting only protein–protein interactions with a confidence level greater than 0.7. R package “networkD3” (https://cran.r-project.org/web/packages/networkD3/, accessed on 10 January 2020) software was used for visualizing.

## 3. Results

### 3.1. Protein Identification

In this study, the proteomes of ejaculated spermatozoa from bucks and rams were identified using the TMT-labeled proteomics approach. The raw data have been uploaded to the ProteomeXchange Consortium (http://proteomecentral.proteomexchange.org, accessed on 2 May 2020) via the iProX partner repository, with the dataset identifier PXD024653. Mass errors of identified peptides were concentrated in the range below 5 ppm, and the lengths of most peptides were between 8 and 20 amino acids, consistent with typical lengths of trypsin-digested peptides; these indicated that that samples met the standard (Appendix A, Appendix A). The stability of the data was evaluated to examine the variation in abundance among samples in the same group; the reproducibility test was shown by calculation of the Pearson correlation coefficient (Appendix A). Principal component analysis (PCA) for the quantification of buck and ram sperm proteins showed that the samples were in separate clusters (Figure 2). PC1 (94.1%) and PC2 (2.5%) accounted for 96.6% of the total variance, indicating that it was possible to significantly distinguish between the two species of the samples.

After MS analysis, a total of 3127 proteins were identified, of which 2644 were quantified (Appendix A); 279 were filtered to be differentially abundant (FDR < 1%, FC > 2.0 or <0.5, and *p*-value < 0.05), with 153 upregulated and 126 downregulated in buck-ejaculated spermatozoa (Figure 3A, Appendix A). Among the DAPs, the seminal plasma protein PDC-109 showed the highest relative upregulation, and the Solute carrier organic anion transporter family member (SLCO6A1) showed the highest relative downregulation. The top 20 upregulated and downregulated proteins in the bucks compared to the rams are provided in Table 1 and Table 2, respectively.

The subcellular localizations of these DAPs were predicted and classified. The upregulated proteins in buck-ejaculated spermatozoa were concentrated mostly in the cytoplasm (26.8%), followed by the extracellular matrix (22.2%), nucleus (19.6%) and mitochondria (17%) (Figure 3B); meanwhile, the downregulated proteins were concentrated mostly in the cytoplasm (27.8%), followed by the extracellular matrix (23.8%), mitochondria (18.3%) and nucleus (11.1%) (Figure 3C). Identifying the overall subcellular localization of the DAPs will help understand their functions and provide a reference for further studies.

### 3.2. Cluster Analysis

The abundance intensities of the identified DAPs were subjected to cluster analysis (Figure 4) and unsupervised hierarchical clustering based on the R language. The results suggested that the DAPs in the spermatozoa ejaculated from bucks and rams were clustered into two different classes. Significant differences in protein abundance intensity were observed between the bucks and rams groups, indicating differences in protein abundance levels between the two groups.

### 3.3. Functional Enrichment Analysis

In order to investigate the potential effects of DAPs in buck- vs. ram-ejaculated sperm, we performed GO terms and KEGG pathway analysis, and the top 20 enriched GO terms are shown in Figure 5. The upregulated proteins were mostly enriched in sperm motility, flagellated sperm motility, cellular lipid catabolism, microtubule bundle formation, fatty acid metabolic and fatty acid catabolic processes (Figure 5A), whereas downregulated proteins were mostly enriched in proteasome-mediated ubiquitin, telencephalon development and glutamate metabolic processes (Figure 5B). The upregulated proteins are enriched in pathways including fatty acid degradation, valine, leucine and isoleucine degradation, and complement and coagulation cascades (Figure 6A); meanwhile, the downregulated proteins were mostly enriched in proteasome, alanine, aspartate and glutamate metabolism, and DNA replication (Figure 6B). Functional enrichment of these DAPs indicated spermatozoa differences of the two species (detailed results are described in Appendix A).

### 3.4. Proteins Network Analysis

STRING analysis of DAPs from buck- versus ram-ejaculated spermatozoa formulated a color-coded network that was largely based on their association [21]. Partial DAPs such as HSP90AA1, ACLY, PSMD13, ACACB, LDHC, PSMD4, PSMD12 and PSMD11 acted as “cross-talk” nodes in the functional modules as key intermediates or enzymes, which were mainly involved in response to stimuli, catalytic activity, molecular function regulation, membrane, metabolism and single-organism processes, macromolecular complex and others (Figure 7, Appendix A).

### 3.5. Western Blot

To verify the DAPs identified with the TMT LC–MS/MS analysis, five differential proteins (PSMD3, DNAJB13, ACLY, DLST and HSP90AA1) were selected to validate the proteomic data using Western blotting (Figure 8A). From Figure 8B,C, it can be found that the expression amounts and expression profiles of these five proteins are consistent with the TMT results, indicating that the proteomic data obtained by TMT technology in this study are solid.

## 4. Discussion

### 4.1. Proteome Characterization of the Spermatozoa

Ejaculates undergo a series of basic processes that culminate in the formation of a viable embryo: energization, hyperactivation, penetration of the zona pellucida (ZP) and fusion of the sperm–egg cell membrane [22]. These biological changes are closely related to the expression and configuration of sperm proteins [23]. Proteomic analyses of spermatozoa and seminal plasma are favorable methods to characterize meaningful proteins that are involved in the biological mechanism of sperm functional execution, serving male potential fertility. Using TMT LC–MS/MS, a total of 2644 sperm proteins from bucks and rams were identified and quantified, and 279 DAPs met the selection criteria. More optional proteins were characterized than in our previous study that used DIA–MS proteomics [21]. These DAPs are mostly involved in sperm motility and various amino acid metabolic processes, similar to the functional characteristics of proteins from ram spermatozoa that were identified by GeLC–MS/MS [24]. In addition, Lv et al. identified 39 differential proteins using the 2DE coupled with MS technique, with 28 found to be upregulated and 11 downregulated in fresh versus post-thaw sperm of the Zhaotong ram. These differential proteins were involved in sperm metabolism, motility and ROS levels, which indicate that the cryopreservation process modifies the proteome and alters the longevity of ram sperm [14,25]. Combined with our results, these sperm differential proteins are usually involved in metabolic pathways and alter the quality of sperm, consequently influencing their fertility after natural mating or AI.

### 4.2. Differences in Spermatozoa Proteins between Bucks and Rams

An interspecific comparison revealed significant sperm characteristics between bucks and rams, which were substantiated by the outcomes in PCA. This was confirmed by the results of PCA. There were 153 proteins upregulated and 126 downregulated in buck versus ram spermatozoa, respectively.

Studies using mammalian sperm have uncovered information on the action of the ubiquitin-proteasome system (UPS) in the modulation of fertilization, involving sperm interactions with ZP, as well as early stages of sperm acquisition, the remodelling of the sperm plasma membrane and apical body, and the gain in sperm fertilization capacity [26,27]. Differentially abundant proteins of this study were enriched in the proteasome pathway: PSMA3 (proteasome subunit α type-3), PSMA7 (proteasome subunit α type-7), PSMA8 (proteasome subunit alpha4s), PSMC2 (proteasome 26S subunit ATPase 2), PSMD3 (26S proteasome non-ATPase regulatory subunit 3), PSMD4 (26S proteasome non-ATPase regulatory subunit 4), PSMD11 (26S proteasome non-ATPase regulatory subunit 11) and PSMD14 (26S proteasome non-ATPase regulatory subunit 14). Protein–protein network analyses indicated that these DAPs have a strong interaction in spermatozoa as components of the 26S proteasome. In eukaryotes, the 26S proteasome is a conserved multimeric protease consisting of two complexes: a hollow 20S core subunit and a 19S regulatory subunit; it is primarily used for protein-dependent proteolysis, affecting transcription factors and other intracellular regulatory events [28,29,30]. In mice, the substrates such as PSMA7, an α subunit of the 20S core expresses in the adult testis, is mediated by the zygote-specific proteasome assembly chaperone (ZPAC), and defends germ cell survival during spermatogenesis [31]. PSMA8 is responsible for male fertility by assembling a type of testis-specific 20S core proteasome, expresses in spermatocytes from the pachytene stage and interacts with several key meiotic players (SYCP3, SYCP1, CDK1 and TRIP13), and gives rise to normal round spermatids [32]. PSMA8 deletion delays entry into the M phase by activating phase I proteins; a decrease in PSMA7 stops it at this stage and ultimately causes male infertility [33]. Additionally, PSMD4 is a non-ATPase member of the 19S particle, and recognizes and binds to multi-ubiquitin chains. It is detected in the acrosomal region of zona-bound porcine spermatozoa, and is involved in sperm-ZP penetration during fertilization [34]. In our comparison of bucks versus rams, these subunits were downregulated, which implied their stronger roles in ram fertility.

On the other hand, we also observed that HSP90AA1 (heat shock protein HSP 90-α, HSP90-α), ACLY (ATP-citrate lyase), ACACB (acetyl-coenzyme A carboxylase β), DLST (dihydrolypoamide S-succinyl transferase), SLCO6A1 (solute carrier organic anion transporter family member) were downregulated in buck versus ram spermatozoa, while DNAJB13 (DnaJ heat shock protein family member B13) and P02784 (seminal plasma protein PDC-109) were upregulated in the comparable group. Studies have shown that HSP90AA1 is mainly abundant in the cytoplasm of germ cells from the beginning of the meiotic phase, and is necessary for germ cell proliferation and differentiation [35,36]. This protein also localizes on the plasma membrane of the rat spermatozoa head, and assist in the spermatozoa–oocyte fusion process [37]. In boar spermatozoa, HSP90AA1 level is significantly higher in good frozen ejaculates than in the poor, linking its protective roles in the resistance against cell oxidative stress and apoptosis, hence in sperm motility after cryopreservation [38]. It is inducible and involved in stress-induced cytoprotection via various cellular processes, such as protein folding and degradation, and signal transduction cascades [39].

Cellular energy metabolism is closely linked to cell fate. Optimizing the distribution of tricarboxylic acid (TCA) key enzymes in mitochondria and the cytoplasm increase the activity of the electron transport chain (ETC), which improves oxidative phosphorylation (OXPHOS) to ATP production, resulting in good motility of mammalian spermatozoa [40]. ACLY is the main enzyme responsible for the synthesis of intracellular acetyl coactivators to initiate the TCA cycle; therefore, it provides the necessary electrons for the ETC [41]. In heat-stressed testes, a decrease in ACLY was found to be associated with germ cell apoptosis, and ACLY levels in spermatozoa were recognized as a target of potential heat sensitivity in germ cells [42]. ACACB is a flux-determining enzyme that localizes on the outer mitochondrial membrane; it limits mitochondrial fatty acid oxidation (FAO) by catalyzing the formation of malonyl-CoA from acetyl-CoA to inactivate carnitine-palmitoyl-CoA transferase I [43]. FAO is an important source of energy in the form of ATP, especially when glucose and glycogen stores are low [44]. DLST is one of necessary enzymatic subunits to form the oxoglutarate dehydrogenase complex (α-ketoglutarate dehydrogenase), the complex which converts oxoglutarate to succinyl-CoA in the TCA cycle, and eventually generates energy [45]. Amaral et al. reported a much lower DLST protein abundance in low-motility spermatozoa than in normal spermatozoa [46].

Solute carrier (SLC) transporters are a large family of proteins that are involved in the movement of a wide variety of materials in a wide variety of tissues and cells. The transport directly requires ATP supply [47]. Studies have found that subfamilies such as SLC22 act as organic anion/cation transporters in mice spermatozoa, which relates to fertility [48]. Combined with our finding that there is more abundance of SLCO6A1 (OATP6A1) in ram spermatozoa, this implies it has a transmembrane effect as the gonad-specific anion transporter in spermatozoa. Interestingly, we also found DNAJB13 was significantly enriched. It is a type II HSP40/DnaJ protein that is represented in the spermatid cytoplasm and flagellum, starting from the fourth week of life in mice, as well as in the axoneme and cyclosome of mature spermatozoa [49]. DNAJB13t mutations in human spermatozoa result in reduced expression levels of the DNAJB13t protein, which causes defects in the sperm tail and consequently impairs sperm motility, a phenomenon that is, over time, important for sperm flagellation and sperm motility [50]. The difference in DNAJB13 in the spermatozoa of bucks versus rams suggests that the spermatozoa of two species are different in development and motility. Additionally, we also identified the differential seminal plasma protein PDC-109, which contains two fibronectin type II (FnII) structural domains that aid in sperm capacitation by binding to choline phospholipids on the sperm plasma membrane to induce lipid efflux; this, in turn, allows for successful sperm fertilization. In addition, PDC-109, like HSP90AA1, displays chaperone-like properties that protect other proteins from various types of stress [51], protecting the functional status of sperm from harmful environmental factors, such as freezing and AI, on male reproductive function.

## 5. Conclusions

In summary, comparative proteome profiles of spermatozoa of bucks and rams were studied according to TMT proteomics technology. Functional enrichment revealed that these proteomic DAPs were mainly involved in the mitochondria, the extracellular matrix and the nucleus; they were involved in sperm motility, membrane components, oxidoreductase activity, endopeptidase complex and proteasome-mediated ubiquitin dependent protein catabolism. Some DAPs, such as HSP90AA1, ACLY, PSMA7 and PSMD4, act as “cross-talk” nodes in protein–protein networks as key intermediates or enzymes, which are mainly involved in responses to stimuli, catalytic activity and molecular function regulator pathways that are strictly related to spermatozoa function. Our results imply that there are significant differences between sperm proteins in bucks and rams, which provide insights into the molecular mechanisms of ram spermatozoa function, and also promote efficient spermatozoa utilization links to fertility or specific biotechnologies for bucks and rams.

## Figures and Tables

**Figure 1 genes-14-00973-f001:**
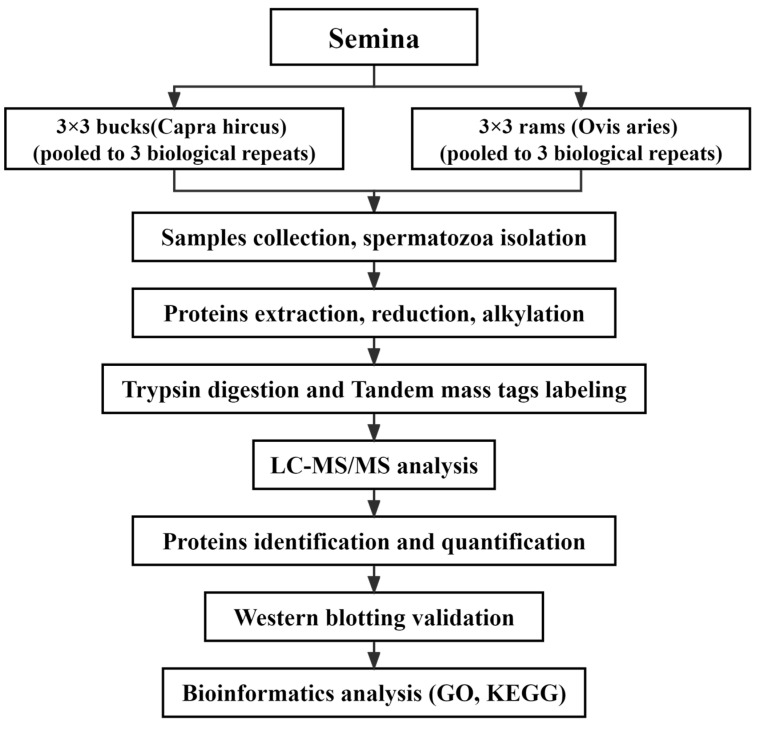
The identification of DAPs between buck sperm and ram sperm. First, high-quality semen were selected from buck and ram ejaculate, then centrifuged to obtain sperm and extract sperm proteins. The disulfide bonds of the proteins were opened with reductive alkylation, modified with sulfhydryl groups and enzymatically cleaved into peptide fragments. The protein mixture was subjected to trypsin digestion and tandem mass tags labeling. Analyses were then carried out using LC–MS/MS. Selected DAPs were validated for protein via Western blotting. Finally, bioinformatics analysis of the data was performed.

**Figure 2 genes-14-00973-f002:**
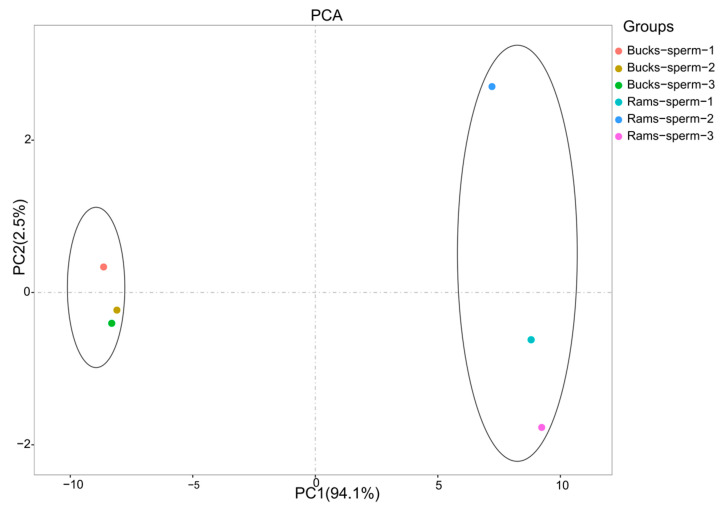
Principal component analysis (PCA) scores plot of qualified spermatozoa proteins from bucks and rams.

**Figure 3 genes-14-00973-f003:**
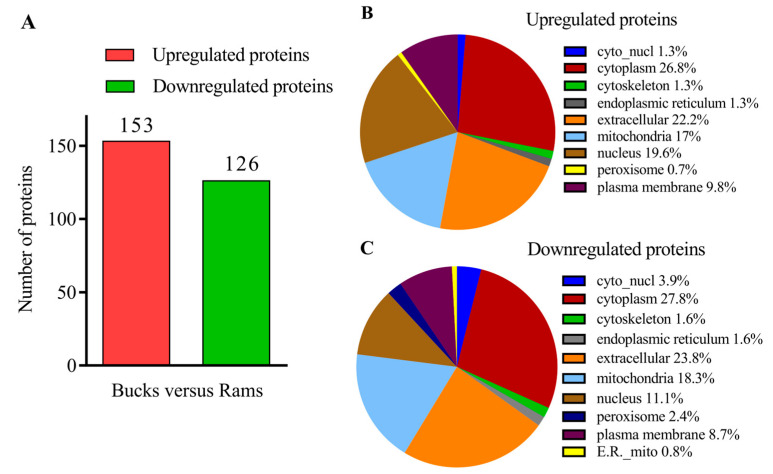
Screening of the DAPs. (**A**) Bar charts of the numbers of DAPs counted. (**B**) Pie charts of the proportions of upregulated proteins at different subcellular locations. (**C**) Pie charts of the proportions of downregulated proteins at different subcellular locations.

**Figure 4 genes-14-00973-f004:**
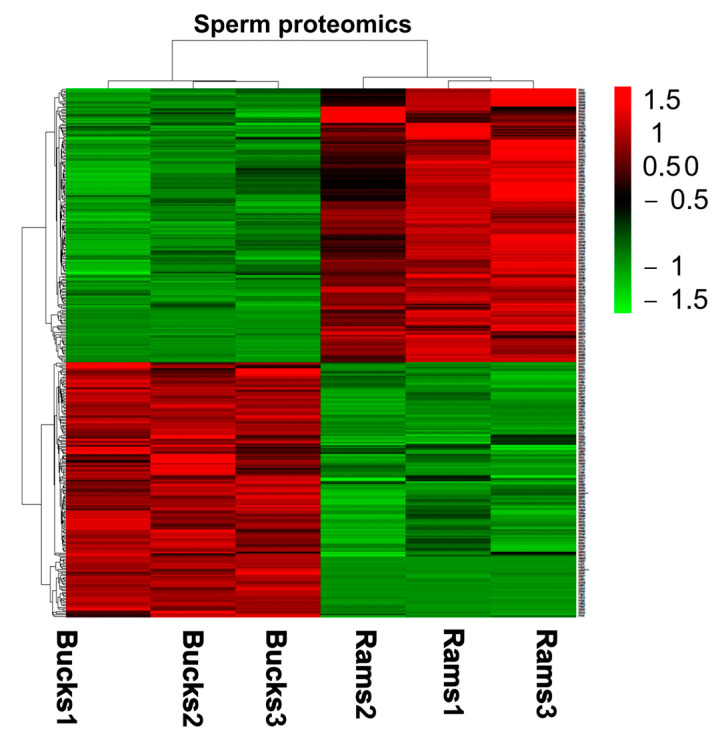
Cluster heat map of expression intensities for the DAPs. Red: expression of upregulated proteins; Green: expression of downregulated proteins; Bucks: bucks spermatozoa; Rams: rams spermatozoa. The numbers 1, 2 and 3 indicate biological repeats in the group.

**Figure 5 genes-14-00973-f005:**
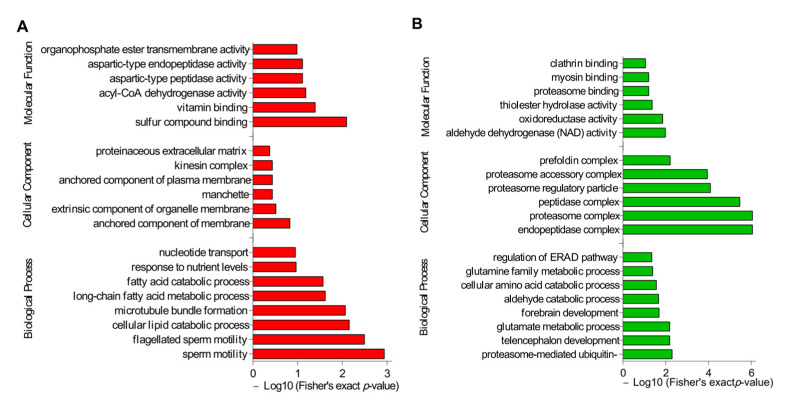
GO annotation analysis of DAPs in the comparison of buck versus ram spermatozoa. (**A**) Red bars show the degree of significance of upregulated proteins, and (**B**) green bars show the degree of significance of downregulated proteins.

**Figure 6 genes-14-00973-f006:**
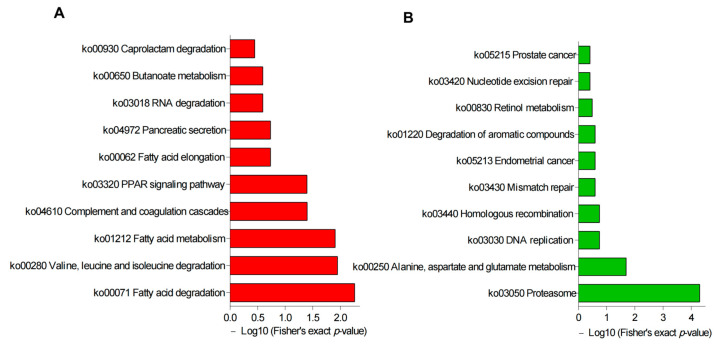
KEGG pathway enrichment analysis results in buck versus ram spermatozoa. (**A**) Red bars represent the degree of significance of upregulated protein enrichment. (**B**) Green bars represent the degree of significance of downregulated protein enrichment.

**Figure 7 genes-14-00973-f007:**
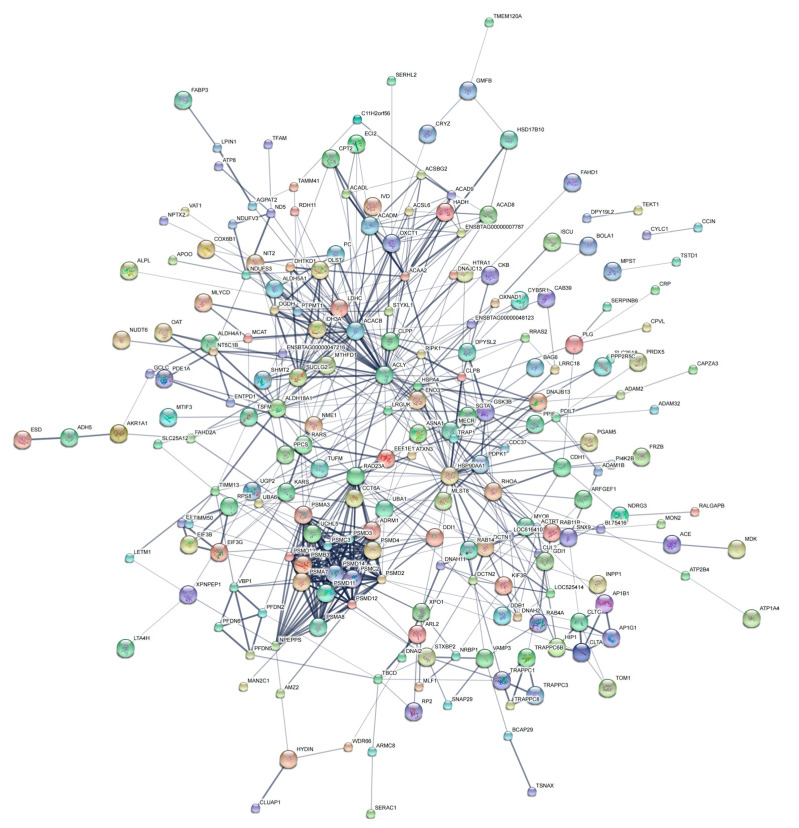
Protein interaction network association map of DAPs in buck and ram sperm. Networks generated using STRING (v.10.5). Only interactions associated with proteins in the identified dataset were selected, excluding external candidates.

**Figure 8 genes-14-00973-f008:**
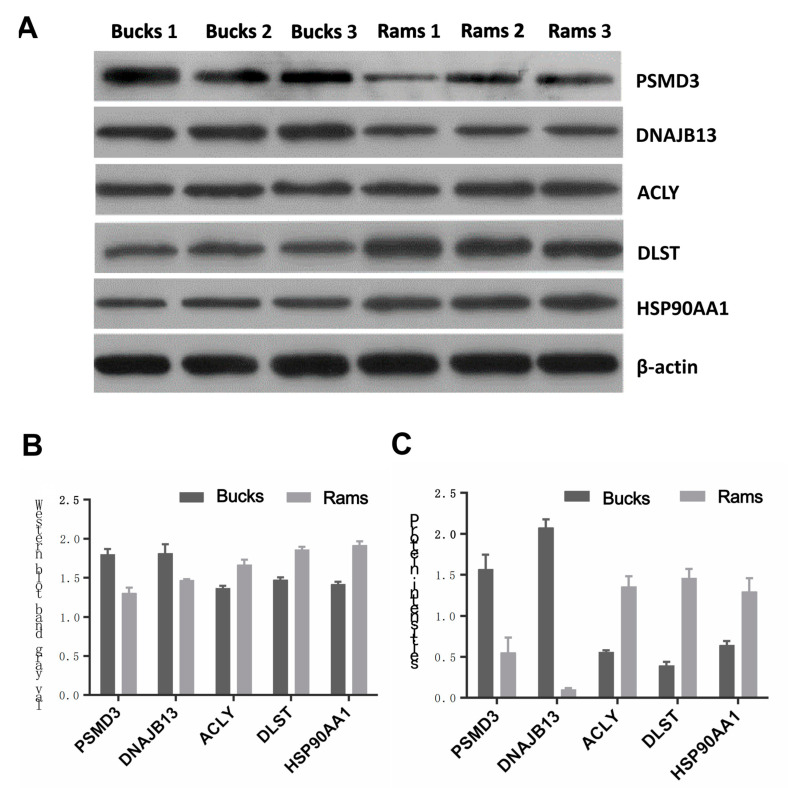
Validation of five selected proteins in bull and ram spermatozoa via Western blotting. (**A**) The abundance of two upregulated (PSMD3, DNAJB13) and three downregulated (ACLY, DLST, HSP90AA1) proteins were analyzed. β-actin protein was used as an internal reference. (**B**) Histogram of quantitative Western blotting results for the five proteins. (**C**) Histogram of quantitative results of TMT analysis for the five proteins. All values are expressed as mean ± SEM (n = 3 per group), and the differences in tests are statistically significant (*p* < 0.05).

**Table 1 genes-14-00973-t001:** The top 20 proteins that were more abundant in bucks compared to rams.

Accession	Description	Gene Name	MW [kDa]	Peptides	Coverage	Accession	Description	Gene Name
P02784	Seminal plasma protein PDC-109		15.48	1	5.2	1	0.002299	47.32
Q2TA26	Coiled-coil domain-containing 116	CCDC116	40.31	3	14.3	1	0.000542	27.11
F1MB31	PREDICTED: putative serine protease 46	PRSS46	33.36	2	7.3	1	0.001534	26.54
O77780	Disintegrin and metalloproteinase domain-containing protein 2	ADAM2	83.15	6	5.8	2	9.23 × 10^−7^	25.22
E1BPJ2	Coiled-coil domain-containing protein 8	CCDC81	76.13	23	31.9	2	0.001281	22.90
Q02368	NADH dehydrogenase [ubiquinone] 1 β subcomplex subunit 7	NDUFB7	16.40	4	40.1	1	0.000444	22.56
Q3SZW9	DnaJ (Hsp40) related, subfamily B, member 13	DNAJB13	36.08	14	47.2	2	0.001758	21.13
Q32PA1	CD59 molecule, complement regulatory protein	CD59	13.66	1	7.4	1	0.017282	21.03
E1B715	Kinesin-like protein	KIF9	89.82	16	22.9	1	0.009728	19.94
G3N1S7	Cilia- and flagella-associated protein 44 isoform X1		115.08	19	22.8	1	0.020836	16.25
F1MI34	Coiled-coil domain-containing protein 108-like isoform X1	CFAP65	206.06	14	8.8	1	1.46 × 10^−5^	14.65
G3MWG7	EF-hand calcium-binding domain-containing protein 6 isoform X1	EFCAB6	173	25	20.7	3	7.76 × 10^−5^	13.56
Q3SZ00	HADHA protein	HADHA	83.25	26	38.1	1	0.002899	13.38
Q32KP0	Spermatid-specific manchette-related protein 1	SMRP1	35.07	15	47	1	0.000412	13.33
Q32KS3	“Capping protein (Actin filament) muscle Z-line, α 3	CAPZA3	35.07	12	38.5	2	0.00056	12.96
A6QPW2	MGC157332 protein	MGC157332	30.57	9	26.2	2	0.001011	11.99
Q58D55	β-galactosidase	GLB1	73.41	4	7.8	1	0.007557	11.59
W5P004	Midkine isoform X1	MDK	15.57	9	49	9	0.016649	11.17
Q2TA11	Uncharacterized protein C1orf158 homolog		23.23	5	29.1	1	5.54 × 10^−5^	10.83
F1MJM3	Disintegrin and metalloproteinase domain-containing protein 20	ADAM20	79.58	4	5.2	2	0.000432	10.47

**Table 2 genes-14-00973-t002:** The top 20 proteins that were more abundant in rams compared to bucks.

Accession	Description	Gene Name	MW [kDa]	Peptides	Coverage	Accession	Description	Gene Name
W5QAF5	Solute carrier organic anion transporter family member	SLCO6A1	79.53	3	5.2	3	0.018435	17.20
W5QIT9	LOW QUALITY PROTEIN: phospholipase DDHD1 isoform X1	DDHD1	98.70	4	4.1	1	0.043971	16.45
W5NPV3	Disintegrin and metalloproteinase domain-containing protein 5-like isoform X1		21.37	3	12.1	3	0.044157	13.66
W5Q563	GDP-L-fucose synthase isoform X1	TSTA3	36.03	1	4.4	1	0.013513	11.82
W5PZ19	Equatorin isoform X1	EQTN	33.80	4	12	2	0.001924	11.07
W5PLD6	Sialomucin core protein 24 isoform X1	CD164	20.59	1	6.6	1	0.020619	10.31
W5NTS0	Lymphocyte antigen 6K	LY6K	17.83	2	12.5	2	0.033376	10.26
W5QI97	Cytoskeleton-associated protein 2-like isoform X2	CKAP2L	85.60	1	1.3	1	0.029717	9.94
W5PT91	Ubiquitin-conjugating enzyme E2 J1 isoform X4	UBE2J1	35.68	4	15.4	1	0.003098	9.61
W5PD71	Pentaxin	CRP	25.27	7	33.5	7	0.02044	9.50
W5Q4D9	NAD(P)(+)--arginine ADP-ribosyltransferase	ART3	43.75	11	22.7	7	0.022189	9.23
W5PNP1	Lactadherin isoform X1	MFGE8	48.36	14	36.1	3	0.014737	8.91
W5PSI1	26S proteasome non-ATPase regulatory subunit 3 isoform X2	PSMD3	61.66	13	22.6	1	0.034579	8.09
W5QET6	Monocarboxylate transporter 1	SLC16A1	54.19	8	16.2	1	0.004748	7.49
W5PU33	T-complex protein 1 subunit zeta isoform X1		44.13	5	11.2	2	0.014637	7.33
W5QIQ0	C-type lectin domain family 2 member F-like isoform X1	LOC101120482	22.34	3	16.7	3	0.004093	7.03
F1MNL6	Nuclear pore membrane glycoprotein 210-like	NUP210L	206.8	30	17	1	0.006081	6.91
W5Q104	Disintegrin and metalloproteinase domain-containing protein 20-like	LOC101111942	83.46	8	11	6	0.005871	6.78
E1BAR0	Small integral membrane protein 5	SMIM5	8.76	1	19.2	1	0.002139	6.77
W5Q0G6	Serine protease 55	PRSS55	35.99	6	29.9	6	0.012645	6.73

## Data Availability

All of the data generated and analyzed during this study are included in this published article. The raw data supporting the findings of this study are available from the corresponding author (zhangzijun@ahau.edu.cn) on request.

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
