# Peer review of "TMT-Based Comparative Proteomic Analysis of the Spermatozoa of Buck (Capra hircus) and Ram (Ovis aries)"

_genes, 2023, doi:10.3390/genes14050973_

Round 1
Reviewer 1 Report
The manuscript genes-2287366, entitled “TMT-Based Comparative Proteomic Analysis of Goat and Sheep Spermatozoa”, is an elemental research to understand physiological and biochemical changes in the male genital tract, in particular, with the functional proteins involved in cellular regulation mechanism into the sperm fertility. This manuscript is well written. All methodology about protein extraction and its identification was well done as well as bioinformatics analysis. The number of bucks and rams was appropriate for the study. The authors must attend to some recommendations:
In all document, use the words bucks and rams instead goat and sheep, respectively.
Line 234: Write or put the name for the Table 2.
Line 442: It is necessary to include Data Availability Statement: Data are available from …. author e-mail.
Author Response
Dear Reviewer/Editor:
Thank you for your letter and the reviewers’ comments concerning about our manuscript “TMT-Based Comparative Proteomic Analysis of Bucks and Rams Spermatozoa” (genes-2287366). Those comments are all valuable and very helpful for revising and improving our manuscript. We have substantially revised the manuscript according to your comments and suggestions, all details as follows.
Thanks a lot again!
Reviewer’s Comments and Suggestions:
The manuscript genes-2287366, entitled “TMT-Based Comparative Proteomic Analysis of Goat and Sheep Spermatozoa”, is an elemental research to understand physiological and biochemical changes in the male genital tract, in particular, with the functional proteins involved in cellular regulation mechanism into the sperm fertility. This manuscript is well written. All methodology about protein extraction and its identification was well done as well as bioinformatics analysis. The number of bucks and rams was appropriate for the study. The authors must attend to some recommendations:
In all document, use the words bucks and rams instead goat and sheep, respectively.
Line 234: Write or put the name for the Table 2.
Line 442: It is necessary to include Data Availability Statement: Data are available from …. author e-mail.
Responds to Reviewer’s Comments and Suggestions:
Point 1. In all document, use the words bucks and rams instead goat and sheep, respectively.
Response: Thank you for your suggestions, which are very important for the improvement of our manuscript. According to your suggestion, we have use the words bucks and rams instead goat and sheep in the revised manuscript.
Point 2. Line 234: Write or put the name for the Table 2..
Response: Thanks for your reminder. We have amended it in line 257.
Point 3. Line 442: It is necessary to include Data Availability Statement: Data are available from …. author e-mail.
Response: Thank you for your suggestions, we have added Data Availability Statement in line 473 - 475.

Reviewer 2 Report
It is suggested to the authors to describe if the feeding of the experimental animals was based on only forage. Indicate the scientific name of the forage and the amount offered to feed the males during the study.
Author Response
Dear Reviewer/Editor:
Thank you for your letter and the reviewers’ comments concerning about our manuscript “TMT-Based Comparative Proteomic Analysis of Bucks and Rams Spermatozoa” (genes-2287366). Those comments are all valuable and very helpful for revising and improving our manuscript. We have substantially revised the manuscript according to your comments and suggestions, all details as follows.
Thanks a lot again!
Reviewer’s Comments and Suggestions:
It is suggested to the authors to describe if the feeding of the experimental animals was based on only forage. Indicate the scientific name of the forage and the amount offered to feed the males during the study.
Responds to Reviewer’s Comments and Suggestions:
Point 1. It is suggested to the authors to describe if the feeding of the experimental animals was based on only forage. Indicate the scientific name of the forage and the amount offered to feed the males during the study.
Response: Thank you for your suggestions, which are very important for the improvement of our manuscript. During the course of this study, our animals were fed in strict accordance with the company's regulations and animal welfare requirements. Each animals was fed 2 kg fresh grass, 0.8 kg hay and 0.5 kg mixed concentrate (half maize and half soya meal) per day during the period of semen collection. Also in this manuscript revision, we have added a description of the feeding of experimental animals.

Reviewer 3 Report
I read the paper “ TMT-Based Comparative Proteomic Analysis of Goat and Sheep Spermatozoa” with very attention. This topic in the field of sheep and goat is very important and deserves to be deepened. The study is well developed, but there is, for me, a crucial problem: the low the low number of samples and the statistical analysis not supported by any statistical significance. Therefore, it’ necessary to describe the consistency of Anhui goats and Hu sheep to well clarify the low number of animals and support the statistical analysis with test of significance.
Author Response
Dear Reviewer/Editor:
Thank you for your letter and the reviewers’ comments concerning about our manuscript “TMT-Based Comparative Proteomic Analysis of Bucks and Rams Spermatozoa” (genes-2287366). Those comments are all valuable and very helpful for revising and improving our manuscript. We have substantially revised the manuscript according to your comments and suggestions, all details as follows.
Thanks a lot again!
Reviewer’s Comments and Suggestions:
I read the paper “ TMT-Based Comparative Proteomic Analysis of Goat and Sheep Spermatozoa” with very attention. This topic in the field of sheep and goat is very important and deserves to be deepened. The study is well developed, but there is, for me, a crucial problem: the low the low number of samples and the statistical analysis not supported by any statistical significance. Therefore, it’ necessary to describe the consistency of Anhui goats and Hu sheep to well clarify the low number of animals and support the statistical analysis with test of significance.
Responds to Reviewer’s Comments and Suggestions:
Point 1. The study is well developed, but there is, for me, a crucial problem: the low the low number of samples and the statistical analysis not supported by any statistical significance. Therefore, it’ necessary to describe the consistency of Anhui goats and Hu sheep to well clarify the low number of animals and support the statistical analysis with test of significance.
Response: Firstly, thank you very much for your acknowledgement. In the field of sperm proteomics studies, there are a large number of studies that use n=3 as sample size (Leahy et al, 2020; Pini et al, 2016; Zhu et al, 2020). In addition, in this study we conducted proteomics studies in separate randomly selected bucks and rams populations. The proteomic quality control results indicate that the samples selected for our study were representative.
Leahy, T.; Rickard, J.P.; Pini, T.; Gadella, B.M.; Graaf, S.P. Quantitative Proteomic Analysis of Seminal Plasma, Sperm Membrane Proteins, and Seminal Extracellular Vesicles Suggests Vesicular Mechanisms Aid in the Removal and Addition of Proteins to the Ram Sperm Membrane. Proteomics. 2020 Jun;20(12):e1900289.
Pini, T.; Leahy, T.; Soleilhavoup, C.; Tsikis, G.; Labas, V.; Combes-Soia, L.; Harichaux, G.; Rickard, J. P.; Druart, X.; Graaf, S. P. Proteomic Investigation of Ram Spermatozoa and the Proteins Conferred by Seminal Plasma. Journal of proteome research, 2016;15(10):3700–3711.
Zhu, W.; Zhang, Y.; Ren, C.H.; Cheng, X.; Chen, J.h.; Ge, Z.Y.; Sun, Z.P.; Zhuo, X.; Sun, F.F.; Chen, Y.L.; Jia, X.J.; Zhang, Z.J. Identification of proteomic markers for ram spermatozoa motility using a tandem mass tag (TMT) approach. J Proteomics. 2020, 210, 103438.
